# Can Cisplatin Therapy Be Improved? Pathways That Can Be Targeted

**DOI:** 10.3390/ijms23137241

**Published:** 2022-06-29

**Authors:** Reem Ali, Mustapha Aouida, Abdallah Alhaj Sulaiman, Srinivasan Madhusudan, Dindial Ramotar

**Affiliations:** 1Division of Biological and Biomedical Sciences, College of Health and Life Sciences, Hamad Bin Khalifa University, Education City, Qatar Foundation, Doha P.O. Box 34110, Qatar; maouida@hbku.edu.qa (M.A.); abal36503@hbku.edu.qa (A.A.S.); 2Biodiscovery Institute, School of Medicine, University of Nottingham, University Park, Nottingham NG7 3RD, UK; srinivasan.madhusudan@nottingham.ac.uk

**Keywords:** cisplatin, cisplatin resistance, DNA repair, platinum sensitisation

## Abstract

Cisplatin (*cis*-diamminedichloroplatinum (II)) is the oldest known chemotherapeutic agent. Since the identification of its anti-tumour activity, it earned a remarkable place as a treatment of choice for several cancer types. It remains effective against testicular, bladder, lung, head and neck, ovarian, and other cancers. Cisplatin treatment triggers different cellular responses. However, it exerts its cytotoxic effects by generating inter-strand and intra-strand crosslinks in DNA. Tumour cells often develop tolerance mechanisms by effectively repairing cisplatin-induced DNA lesions or tolerate the damage by adopting translesion DNA synthesis. Cisplatin-associated nephrotoxicity is also a huge challenge for effective therapy. Several preclinical and clinical studies attempted to understand the major limitations associated with cisplatin therapy, and so far, there is no definitive solution. As such, a more comprehensive molecular and genetic profiling of patients is needed to identify those individuals that can benefit from platinum therapy. Additionally, the treatment regimen can be improved by combining cisplatin with certain molecular targeted therapies to achieve a balance between tumour toxicity and tolerance mechanisms. In this review, we discuss the importance of various biological processes that contribute to the resistance of cisplatin and its derivatives. We aim to highlight the processes that can be modulated to suppress cisplatin resistance and provide an insight into the role of uptake transporters in enhancing drug efficacy.

## 1. Background

Cisplatin is one of the most potent anti-cancer agents that is effective against solid tumours. It is also used for treating lymphomas, sarcomas, and germ cell cancers.

Cisplatin was first synthesised by M. Peyrone in 1844 and its chemical structure was reported by Alfred Werner in 1893 [1]. However, it did not receive scientific interest until 1965, when Dr. Rosenberg at Michigan State University discovered that an oxidising agent generated from the electrolysis of platinum mesh electrodes was capable of inhibiting cell division in *Escherichia coli* in a bacterial chamber [2]. The author proposed that this agent will affect actively dividing cancer cells and suggested using these platinum derivatives in cancer chemotherapy. Platinum has a unique structure as a metallic compound with a square planar geometry (Figure 1A). It has a crystalline powder texture of white or deep yellow in colour and is slightly soluble in water and dimethyl primanide and *N*,*N*-dimethylformamide. Cisplatin is stable at room temperature but may transform slowly over time to the *trans*-isomer rendering it clinically ineffective in DNA crosslinking. Cisplatin has a molecular weight of 301.1 gm/mol, a density of 3.74 g/cm^3^, a melting point of 270 °C, a log K_ow_ of −2.19 and a water solubility of 2.53 g/L at 25 °C [3].

Since the identification of cisplatin and its approval by the FDA in 1978, it has proven to have an anti-proliferative effect against different tumour lineages, including sarcomas, bones, muscles, and blood vessels. Although it revolutionised the treatment of solid tumours, a major problem with this agent is the increased nephrotoxicity, which fosters continuous research elucidating other platinum derivatives with equal potency and less toxicity [4].

## 2. Mechanisms of Cisplatin Toxicity

Cisplatin toxicities were identified as early as the discovery of its anti-tumour functions and it includes gastrointestinal toxicity, myelosuppression, neurotoxicity, vascular and ototoxicity. However, the most severe side effect of cisplatin is nephrotoxicity. The severity of the toxicity is dose-dependent which limits the administration of therapeutic doses to patients [5]. Cisplatin is excreted by glomerular filtration and tubular secretion; it accumulates in the renal parenchymal cells in high concentrations leading to acute kidney injury. Cisplatin undergoes biotransformation in the apical surface of renal cells, where it becomes conjugated to the tripeptide glutathione (glutamate, cysteine and glycine) by glutathione-S-transferase. Inside the kidney, cisplatin-glutathione conjugates are targeted by gamma-glutamyl transpeptidase (GGT), which cleaves the cisplatin glutathione conjugates into cysteinyl-glycine-conjugates that are more prone to enter the proximal tubules. There it is further metabolised into potent nephrotoxic thiols by the enzyme cysteine-*S*-conjugate β-lyase.

Earlier work attempted to reduce nephrotoxicity by inhibiting the cisplatin metabolic pathway in the kidney by targeting GGT and cysteine *S*-conjugate β-lyase in mice with acivicin and aminooxy acetic acid (AOAA), respectively [6]. However, it has been shown that GGT is essential for cisplatin detoxification through the interaction with cisplatin metabolites [7]. Amifostine is an FDA-approved nephroprotective drug to minimise cisplatin-associated nephrotoxicity, but the overall benefit of using it with cisplatin is not confirmed [8].

## 3. Cisplatin Derivatives

Other platinum analogues have been developed and investigated in solid tumours as a way to minimise the side effects of cisplatin. Carboplatin was the first attractive platinum derivative in which the two chloride molecules of cisplatin are replaced with cyclobutene-decarboxylate groups and exist in the same planar structure as platinum molecule (Figure 1B) [9]. Carboplatin was believed to have less nephrotoxicity, gastrointestinal and ototoxicity compared to cisplatin [10]. This is due to the slower hydrolysis of decarboxylating ligands in comparison to the labile chloride molecules in the cisplatin structure. However, carboplatin is associated with myelosuppression, predominantly thrombocytopenia which is dose-limiting for carboplatin [11]. The DNA adducts formed by carboplatin are similar to cisplatin, but the cyclobutene-decarboxylate leaving groups require esterase cleavage affecting the rate of adduct formation and calculated to be 10-fold slower compared to cisplatin [9]. As such, 20- to 40-fold higher concentrations of carboplatin are required for effective treatment.

Carboplatin is approved for the treatment of ovarian and testicular cancers [12]. It is also widely used for treating head and neck [13], oesophageal [14], cervical, salivary gland, retinoblastoma [15] and glioblastoma cancers [16]. However, carboplatin is not beneficial for treating cisplatin-resistant tumours as the mechanisms of resistance for both drugs are similar. Carboplatin-resistant cells exhibit alteration in mitotic checkpoints, and the combination of carboplatin and prexasertib (CHEK1 inhibitor) suppressed the growth of carboplatin-resistant triple-negative breast cancers xenografts in mice [17]. Thus, checkpoint inhibitors could prevent the propagation of carboplatin-resistant cancer cells. The availability of nearly 250 antibodies that specifically detect checkpoint proteins that are phosphorylated in response to drug treatment would be useful to pre-screen patients who might be candidates for checkpoint inhibitor therapy in response to cisplatin and its derivatives [18]. Table 1 summarises recent clinical studies involving cisplatin derivatives.

The third generation of platinum derivative developed was oxaliplatin. Its structure differs from cisplatin, in which the amine groups of cisplatin are replaced by diaminocyclohexane (DACH) [28] (Figure 1C). Besides the formation of intra-strand and inter-strand DNA adducts, oxaliplatin exerts higher efficacy than cisplatin through inhibition of protein synthesis and induction of ribosomal biogenesis stress that halts translation machinery and cell division [29]. Interestingly, a recent report compared the effect of both oxaliplatin and cisplatin on DNA damage and nucleolar mechanisms, revealing that oxaliplatin, but not cisplatin, significantly inhibited ribosomal RNA synthesis by Pol I without modulating rRNA processing [30]. Oxaliplatin has earned significant importance in recent clinical trials for the treatment of solid tumours, and as such, the combination of oxaliplatin and 5-fluorouracil has been approved for treating advanced colorectal cancer [28,31].

Nedaplatin is another cisplatin analogue with a different leaving group in which glycolate is bound to platinum through a bidentate ligand (Figure 1D). It has a similar mechanism to cisplatin and carboplatin. However, it causes less nephrotoxicity compared to cisplatin [32]. The main toxicity of nedaplatin is myelosuppression, primarily thrombocytopenia [33]. It has been approved in Japan for the treatment of solid tumours of the lung, ovarian, head and neck [34]. Recent in vitro studies showed that encapsulation of nedaplatin on PEGylated liposomes increased its cytotoxicity and enhanced platinum uptake by cancer cell lines [35].

Picoplatin is another platinum-based chemotherapy that was designed to overcome cisplatin and carboplatin resistance [36] (Figure 1E). Picoplatin accumulation has been shown to be high in the cell, and it reached the nucleus despite the presence of high levels of glutathione, which is known to chelate cisplatin. This is due to the steric hindrance structure around the platinum molecule [9]. The in vitro data for picoplatin showed promising anti-tumour activity and overcoming some cisplatin resistance mechanisms in different cancer cell lines. Additionally, early phase I and phase II clinical trials for picoplatin in non-small cell lung cancer and ovarian cancer were well tolerated, although no overall survival benefit or response rate was confirmed over other cytotoxic drugs [37,38]. Importantly, in the SPEAR phase III study conducted on 401 small cell lung cancer patients [39], there was no overall survival benefit for patients treated with picoplatin, which led to the discontinuation of further phase III trials for this agent [33]. However, picoplatin showed some activity on cisplatin- and carboplatin-resistant tumours, and this led to the development of two novel picoplatin derivatives Pt(Oro)(NH_3_)(2-pic), 1, and Pt(5-FOro)(NH_3_)(2-pic), which were synthesised by joining a fragment of picoplatin to orotate or 5-fluoroorotate bioactive ligands. In vitro experiments of these derivatives showed lower toxicity to normal cells and more potency over cisplatin and picoplatin. However, additional in vivo studies are needed to confirm the clinical benefit of these analogues [40].

Another promising cisplatin derivative in clinical trials is lipoplatin, which is formulated from cisplatin (9%) and lipids (91%). This novel tumour drug delivery has increased efficacy against tumours with less toxicity to normal tissues compared to cisplatin [41,42]. A meta-analysis study comparing the efficacy and safety of lipoplatin over cisplatin in non-small cell lung cancer and head and neck cancers revealed that lipoplatin offered survival benefits and less tissue toxicity than cisplatin. The meta-analysis included data from five clinical trials and 523 patients [26]. Table 2 and Table 3 summarise ongoing clinical trials on cisplatin and its derivatives in different cancer types.

### Platinum Loaded Nanoparticles

Significant research focused on limiting cisplatin and carboplatin side toxicities caused by high dose requirements due to poor cellular uptake of the drug [9]. To enhance the therapeutic efficacy of the platinated drugs, researchers have altered the drug delivery system. For example, carboplatin prodrug complex with Fe_3_O_4_ nanoparticles proved to be more toxic than carboplatin in the ovarian cancer cell lines model [43]. The carboplatin nanoparticles are taken up by the endocytosis process leading to increase drug accumulation that can crosslink with the DNA and cause more toxicity to cancer cells than in normal tissues [43]. Similarly, bovine serum albumin nanoparticles were created that encapsulated carboplatin, which was more potent in cytotoxicity experiments in A2780 ovarian cancer cells with 2-fold lower IC50 compared to carboplatin alone [44]. Others have examined carboplatin liposomal nanoparticles in lung cancer in in vitro environments and showed high drug loading efficiency and retention capability [45]. Thus, improving the delivery system for the platinated drugs is likely to reduce the toxicity of non-targeted tissues significantly.

Indeed, self-assembled nanomedicine for cisplatin and lipoplatin, as well as other chemotherapeutic drugs, is an accelerating field of research that is believed to overcome the major toxicity problem limiting their use. Molecular self-assembly is an interesting process by which molecules exist in a disordered system, either static or dynamic and are set to interact locally without external direction. In static self-assembly, the molecules interact to reach an equilibrium state to reduce their free energy, while in dynamic self-assembly, the pre-existing molecules can self-organise to produce a stable structure [46]. Thus, this can overcome the existing drawback in cisplatin conventional delivery systems, as increased detoxification and high drug loading lead to normal tissue toxicity. Moreover, nanoparticles have a prolonged half-life in vivo, allowing optimum platinum accumulation in solid tumours [47]. Carrier self-assembled nanomedicine are approved for other chemotherapeutic drugs like doxorubicin, irinotecan and paclitaxel [48]. A recent study showed that self-assembled Pt (IV-NPs) from biotin-labelled Pt (IV) prodrug demonstrated specific mitochondrial targeting in cancer cells. The self-assembled drug promoted mitochondrial DNA damage and Pt accumulation by reducing GSH levels in vitro as well as in tumour-bearing animal models [49]. Another designed nanoplatform (PDA-pt-NPs) in which Pt was loaded into polydopamine nanoparticles showed efficient cisplatin release in vivo and optimum anti-cancer activity [50]. While only lipoplatin has made it to clinical trials. It will be interesting to evaluate other self-assembled cisplatin-loaded nanoparticles in a clinical setting.

## 4. Cisplatin-Induced DNA Damage

Cisplatin targets a wide range of cellular components, including membrane phospholipids and thiol-containing peptides [51]. It can bind to some proteins as well as the pre-transcription of RNA. However, DNA remains the most critical target for cisplatin [52]. Once inside the cell, cisplatin exchanges one or two of its chlorine molecules with H_2_O causing different types of irreversible DNA lesions [4]. One of the most critical lesions is the intra-strand crosslink that the platinum atom forms covalently with two guanosine nucleotides (GG) or with adenosine-guanosine nucleotides (AG) [52]. Most of the adducts occur as a result of cisplatin binding to purine bases on the same strand of the DNA double helix (Figure 2). As estimated by DNA renaturation studies and alkaline elution, the 1,2 GG intra-strand adduct is the most abundant cisplatin DNA adduct accounting for nearly 60% of the platinum lesions. The next abundant lesion is 1,3 GG intra-strand adduct and GG inter-strand adduct (ICL), which accounts for 1% of the total platinated lesions. It has been suggested that cisplatin tumour cytotoxicity is based on the formation of ICLs, which can prevent DNA synthesis and RNA transcription elongation, and the mechanisms of resistance to the drug stem from tumour cells exploiting multiple DNA repair pathways to process ICL lesions [53].

### 4.1. Nucleotide Excision Repair Pathway

The nucleotide excision repair pathway (NER) plays a key role in repairing cisplatin-induced DNA adducts [54]. NER can be subdivided into two pathways: the transcription-coupled repair (TCR), which mainly processes lesions at the active transcription site in the DNA, and the global genome repair (GGR), which recognises the damage across the majority of the non-transcribing genome. If the adducts occurred at the active transcription site, the lesions would be recognised by RNA polymerase II followed by stalling of transcription elongation and recruitment of several proteins, including the Cockayne syndrome proteins CSA and CSB, the TFIIH complex with XPB and XPD helicases that will unwind the DNA double helix around the lesion. The lesion is then removed by the XPF/ERCC1 and XPG endonucleases that cleave the damaged strand on either side of the lesion to leave a gap of at least 28 to 20 nucleotides [51,52]. The resulting gap is resynthesised by the DNA polymerase ε. It is noteworthy that in the GGR by the NER pathway, the recognition step is initiated by sensor proteins that scan the genomic DNA for chemical distortions such as cisplatin adducts. The protein complex XPC-HR23B together with XPE first recognises the DNA damage, followed by the recruitment of the TFIIH complex to aid in the repair of the lesion by TCR [51,54]. Thus, defects in any proteins in the NER pathway are likely to alter the resistance to cisplatin.

### 4.2. Mismatch Repair Pathway

The mismatch repair pathway (MMR) plays a role in the recognition of DNA damage induced by cisplatin [55]. The MutSα complex consisting of the MSH2-MSH6 heterodimer can recognise ICLs (1–2 bases) caused by cisplatin. In case of more than two bases mispairing, the MutSb complex (MSH2-MSH3) can be activated. Recognition of the mismatched bases by the MutS complexes helps recruit MutLa (MLH1·PMS2 heterodimer), which, in turn, activates EXO1 excision activity to initiate DNA synthesis by polymerase δ followed by the action of DNA ligase I to seal the nick [56,57].

Although MMR is involved in recognising the cisplatin-generated DNA adducts, it cannot repair the lesions because MMR can only replace mispairing opposite the cisplatin adduct. Eventually, unrepaired cisplatin-induced DNA lesion will lead to the formation of DNA double-strand breaks (DSBs) [58]. However, it seems that activation of MMR towards cisplatin adducts could be a rate-limiting step regulating platinum sensitivity and independently of the canonical MMR pathway [4]. MSH3 is a key component in the recognition of the mismatch bases by both MutSα and MutSβ complexes, and its downregulation was shown to sensitise colorectal cancer cells to cisplatin, oxaliplatin, and poly (ADP-ribose) polymerase (PARP) inhibitors [57].

Hypermethylation of the CpG islands of the MLH1 promoter can lead to the accumulation of mismatched nucleotides and the generation of microsatellite instability [59] which have been linked to the poor prognosis of several tumour types but predominantly in colorectal cancers and endometrial cancers [60,61]. Epigenetic silencing of *MLH1* is shown to be associated with aggressive tumours and cisplatin resistance in endometrial cancer [62,63] as well as in testicular germ cell tumours [64,65].

## 5. Role of DSB Pathways in Repairing Cisplatin-Induced DNA Lesions

While cisplatin DNA adducts are more likely to be sensed and processed by single-strand break repair mechanisms such as NER and MMR, these lesions can also trigger double-strand breaks (DSB) [4]. At an active transcriptional site, the TCR pathway can attempt to repair the damage; however, during the DNA incision steps, double-strand breaks could be produced [66]. Nonetheless, cisplatin DNA damage does not appear to trigger an efficient DSB response. Cisplatin combined with ionising radiation is sensed by the Non-Homologous End-Joining (NHEJ) repair pathway [67]. Cisplatin treatment prior to or concurrent with ionising radiation therapies increased cancer cells radiosensitisation in ovarian, head and neck, as well as cervical carcinomas [67]. The exact mechanism for this synergistic effect is not very clear, although it seems to be dependent on the platinum dosage as well as the combination regimen that creates multiple types of DNA lesions. Moreover, cisplatin lesions can inhibit DNA-PK activation lowing its kinase activity and preventing it from binding to the Ku70/80 heterodimer in the NHEJ pathway. In addition, the rate of Ku translocation at the DNA double-strand break is also inhibited in the presence of cisplatin adducts [68]. These findings implicate that the synergism activity between cisplatin and ionising radiation is dependent on cisplatin adducts impairing the NHEJ processing from processing the radiation-induced DNA damage [69,70].

Similarly, cisplatin maximum toxicity is observed in NER deficient tumours such as testicular cancers that have lower expression of XPA and ERCC1, which are key drivers in the NER pathway [71]. In vitro depletion of XPF and ERCC1 sensitised NSCLC to cisplatin [72]. Thus, optimal cisplatin anti-tumour activity could be achieved in patients with known DNA repair defects.

### BRCA Mutations and Cisplatin Sensitivity

Mutations in the breast cancer susceptibility genes *BRCA1* and *BRCA2* confer a high predisposition to breast cancers and other tumour types, including ovarian, pancreatic, and colorectal [73]. BRCA1 colocalises with BARD1, RAD51, and the proliferating cell nuclear antigen (PCNA) during replication. It promotes the 5′ to 3′ resection of the break site leaving behind a 3′ overhang during the S-phase of the cell cycle, whereas BRCA2 binds to RAD51 and regulates RAD51 filament formation. *BRCA1*/*BRCA2* mutations lead to genomic instability because of stalling replication fork during DNA double-strand break repair [74]. A recent report highlighted that ssDNA replication gaps rather than defects in homologous recombination underlie the hypersensitivity of BRCA-deficient cancer to chemotherapy [75]. BRCA1/2 mutated tumours are hypersensitive to ionising radiation and other DNA damaging agents, including platinum derivatives. Breast and ovarian cancers with BRCA mutations are particularly sensitive to platinum. Thus, cisplatin treatment of these tumours showed a high success rate. However, acquired resistance to cisplatin often occurs in recurrent tumours [76,77]. A well-known resistance mechanism occurs through secondary intragenic mutations in BRCA1/2 that restore the wild-type open reading frame and recover homologous recombination [78]. Interestingly, refractory tumours due to BRCA reactivation are as also resistant to the PARP inhibitor olaparib as a monotherapy [79]. Implying similarity in the resistant mechanisms to cisplatin and PARP inhibitors. Loss of MED12, a component of the mediator transcription regulation complex, promotes resistance of BRCA 1/2 deficient cells to cisplatin and PARP inhibitors. MED12 depletion activates the TGF-β pathway independently of the mediator complex and restores HR repair to mediate cisplatin resistance [80].

Emerging studies identify biomarkers of cisplatin resistance and aim to restore tumour sensitivity to platinum derivatives. Mutations in the ATM, RB1 and FANCC genes in muscle-invasive bladder cancer patients (MIBC) correlate with complete response to cisplatin-based neoadjuvant chemotherapy in patients from the clinical trials NCT01031420 and NCT01611662 [81]. In addition, we have previously shown that DNA polymerase beta (Polβ) depletion exquisitely sensitises ovarian cancer cells to cisplatin [82]. We also showed that ovarian tumours with high Polβ expression have poor survival compared to low Polβ expressing tumours. Moreover, we [83] and others [84,85,86] have demonstrated that ERCC1 is a key predictor of cisplatin resistance in ovarian, testicular and lung cancers. Depletion of ERCC1 substantially increased platinum sensitivity in the cell line model [87]. These observations imply that targeting defective DNA repair pathways is a promising approach to modulating cisplatin response.

The Fanconi Anemia pathway (FA) proteins also play a major role in interstrand crosslinks (ICL) repair. ICLs that occur during the S-phase block the replication fork. These ICLs are sensed by the Fanconi Anemia proteins that generate DSBs, which are processed by the homologous recombination pathway. In contrast, the replication-independent ICLs are sensed and processed primarily by the NER pathway. Thus, FA mediated-repair is thought to be directly involved in repairing cisplatin lesions and targeting FA proteins could be a viable approach to modulating cisplatin resistance. Up-regulation of the FA-associated genes, FANCL and RAD18, have been observed in cisplatin-resistant NSCLC cells. Down-regulation of these genes restored platinum sensitisation [88]. A similar approach showed re-sensitisation of the cisplatin-resistant A549/DDP using RNAi for FANCL, FANCD2 or FANCF and inhibition of the FA pathway. A recent study reported that indirect targeting of the FA pathway in squamous cell carcinoma sensitises these tumour cells to cisplatin. Small molecule inhibitor of the deubiquitylase USP28, which is recruited to the DNA damage site in cisplatin-treated cells, indirectly down-regulates FA activation and increases sensitivity to cisplatin [89]. Although the in vitro studies support the inhibition of FA proteins to reverse cisplatin resistance, the clinical development of specific inhibitors has been slow. Early work described several inhibitors of the proteasome machinery as cathepsin B, lysosome, and CHK1 to overcome cisplatin-induced FANCD2 foci formation and FA pathway activation [90]. However, developing more specific inhibitors for key FA proteins will be essential to conducting comprehensive in vivo studies.

## 6. Cisplatin and Apoptosis

The tumour suppressor gene Tp53 bridges cisplatin-induced DNA adducts to apoptosis signalling. Tp53 regulates the signalling of a plethora of cell cycle progression and apoptosis effector genes and plays a prominent role in the cellular response to DNA damage [4]. Upon cisplatin treatment, the kinases ATM and ATR phosphorylate Tp53 on serine 20, leading to its stabilisation. In breast cell lines, cisplatin mediates Tp53 signalling of the BCL-2 pro-apoptotic activator. Additionally, it mediates the expression of the BH-3 only protein Noxa through the end products of lipid peroxidation [91]. Thus, it appears that platinum cellular response can activate different defence mechanisms that interplay to decide the cell fate [92].

## 7. Mechanisms of Cisplatin Resistance

Despite the irreversible DNA damage induced by cisplatin, the development of resistance is usually inevitable. Most patients relapse after the initial response to platinum cycles and this is attributed to the development of one or more resistance mechanisms, which is often a multifactorial process involving intrinsic pathways [51,52]. One important mechanism in cisplatin resistance is the reduction in cellular accumulation of the drug, hence, lowering the levels of platinated DNA adducts [93]. One possible explanation for this could be increased cellular efflux to the drug. Some studies revealed the involvement of the copper transporters ATP7A and ATP7B, responsible for copper detoxification, play a role in cisplatin efflux from the cells [94,95]. Previous work has linked the high expression of ATP7A to cisplatin resistance in the lung [96], oesophagus [97], and ovarian cancer patients. Moreover, preclinical studies showed that overexpression of ATP7A in cervical cancer cells leads to platinum resistance [96]. Another key transporter in platinum detoxification is the multidrug resistance-associated protein MRP1, which functions to efflux different anti-tumour drugs mediating cellular resistance. Currently, several members of the ATP-binding cassette (ABC) are identified as crucial transporters in chemoprotection [52].

## 8. Cisplatin and Immune Response

Cisplatin cell killing mechanisms are not solely attributed to its DNA crosslinking ability but also the ability to interfere with the immune response activation [98]. One suggested mechanism is through induction of the major histocompatibility class complex, MHC class I, which is essential for priming cytotoxic T cells for tumour recognition [99,100]. Early studies reported that cisplatin and vinorelbine doublet upregulate MHC I in lung cancer cell lines [101]. Lung cell lines treated with cisplatin upregulated tumour necrosis factor-α, IL8, CXCL5, and B cell lymphoma-2–like genes (BCL-2) [102]. Similarly, chemotherapeutic drugs, including cisplatin, stimulated MHC I expression through increased interferon-beta signalling in breast cancer cells [103]. More clinical evidence was demonstrated in a study that assessed samples from NSCLC patients treated with cisplatin following radical surgery, showing that 30% of patients had high expression of MHC class I chain-related molecules A and B [99]. Cisplatin-induced MHC class I upregulation was associated with progression-free survival and better prognosis for patients [99].

More promising findings on cisplatin immune response induction are derived from cisplatin combination therapy. The Epitopes-HPV01 and HPV02 trials which investigated the benefit of adding docetaxel to cisplatin plus 5-fluorouracil (DCF) in anal squamous cell carcinoma, found an increase in circulating TH1 T-cells in patients who received the DCF regimen [103,104]. Low levels of myeloid-derived suppressive cells (MDSC) in patients exposed to the DCF treatment were associated with induction of adaptive immune response to hTERT tumour antigen as well as good prognosis [105,106,107]. Combination therapy of low cisplatin dose plus paclitaxel was more effective in immune response induction compared to the maximum tolerated dose of cisplatin in ovarian cancer patients. The low cisplatin dose plus paclitaxel increased IL-2 and IFN-γ associated with cytotoxic CD8(+) T-cell activity [108]. The priming effect of cisplatin has been investigated in combination with a programmed cell death inhibitor (PDL-1). In the ovarian cancer syngeneic mice model, cisplatin plus PDL-1 inhibitor increased CD8+ T-cells and led to tumour regression [109].

Platinum analogues combinations with checkpoint inhibitors PD-1/PD-L-1 have also proved to be promising regimens for suppressing tumour growth, as reported by several investigators [100,110,111]. Wu et al. analysed cisplatin combination with anti-PD-1 antibody (Tislelizumab or Sintilimab) as first-line therapy from the clinical trials (NCT03469557, NCT03748134) [112]. A sublethal dose of cisplatin in oesophageal squamous cell carcinoma (ESCC) induced PD-L1 expression and synergised with an anti-PD-1 antibody [112]. However, this treatment regimen may be limited to certain tumour types. In the Lewis lung carcinoma model, oxaliplatin combination with anti-PD-L1 induced ICD through activation of CD80^+^ CD86^+^ dendritic cells and enhanced cytotoxic T cells (CD8+), resulting in tumour regression [113].

Another well-described mechanism is the ability of platinum derivatives to induce immunogenic cell death (ICD), characterised by the relapse of pre-apoptotic calreticulin and the post-apoptotic high-mobility group box 1 protein (HMGB1) [98,114]. Many chemotherapeutic drugs are known to act by damaging the DNA, followed by secondary processes that involve plasma membrane rupture and release of intracellular content during cell death. It is believed that this process can lead to protein expression at the cell surface, as well as cytokine secretion that could trigger an immune response against tumour cells [115,116]. Oxaliplatin appears to have superior efficiency in the induction of ICD compared to other platinum derivatives [98]. In immunocompetent mice bearing CT26 colorectal cancer cells, both oxaliplatin and cisplatin-induced immune response, but not when CRT was inhibited or depleted or when the toll-like receptor 4 (TLR4) was knocked out [117].

Patients from the LARC study (NCT00278694) with locally advanced rectal cancer and who received a full dose of oxaliplatin induction, followed by an adapted chemoradiation regimen dose, showed a significant increase in HMGB1 during the induction course [118]. HMGB1 was used as a biomarker of ICD and positively correlated with metastasis-free disease. Emerging in vivo studies supports the notion that oxaliplatin is a promotor of CD8+ and CD4+ T-cell response [113,117,119,120]. An increase in CD4+ T-cells activates dendritic cells and other antigen-presenting cells leading to immune response activation. Circulating cytotoxic T-cells implies tumour antigen presentation, activation of an immune cascade of pro-inflammatory cytokines, and type I interferon activation. Haung and colleagues have also found activation of the immune cascade following oxaliplatin and PD-L1 antibody treatment in CT26 colorectal tumours in mice [119].

Elevated levels of pro-inflammatory cytokines such as CCL2, CXCL12, and CXCL13, as well as CXCL9 and CXCL10, which favour T-cell infiltration into the tumour, were found post combination treatment. In addition, their results illustrated activation of Th1-type cytokines IFN-γ and TNF-α, besides IL-4 and IL-10, which are surrogate markers of Th2 response that could promote tumour tolerance [119]. Thus, immune response activation by oxaliplatin checkpoint inhibitor combination is evident. However, harnessing this response through optimising therapeutic doses and regimens is necessary to achieve benefits.

## 9. Cisplatin Uptake Transporters

Several membrane transporters have been proposed to carry cisplatin into the cells through passive diffusion, including Na^+^, K^+^-ATPase, and the solute carriers SLC family of transporters. SLC22A2 (OCT2) and CTR1 are the most described members of the SLC involved in platinum uptake, and OCT2 plays a role in cisplatin transport into renal cells [121]. Mice deleted for OCT1 and OCT2 showed no ototoxicity and mild nephrotoxicity upon cisplatin treatment compared to wild-type mice [122]. Additionally, administration of the OCT2 inhibitor cimetidine with cisplatin in wild-type mice was effective in nephroprotection and significantly reduced cisplatin renal uptake [122]. Another study illustrated that cimetidine reduces acute kidney injury without affecting cisplatin anti-tumour activities [123], indicating the significance of cisplatin transport mechanisms in overcoming toxicities.

The role of copper transporters in the mechanisms of cisplatin resistance is well described, and CTR1 is the primary platinum uptake transporter [97,124,125]. Cisplatin interacts with CTR1 resulting in conformational changes in its methionine residues, allowing the formation of a smaller CTR1 intermediate to promote the drug uptake. However, cisplatin can trigger the degradation of CTR1 and reduce its cellular uptake, thereby promoting resistance to the drug [93].

Importantly, in serous epithelial ovarian cancer patients who received post-operative platinum therapy, high CTR1 mRNA levels were observed in resistant tumours, although the CTR1 protein level was not determined [124,126]. In contrast, NSCLC patients with low CTR1 expression had less intracellular platinum accumulation and poor response [127]. In another study of 54 patients of stage III NSCLC, high CTR1 expression correlated with longer progression-free survival and overall survival [128].

Another copper transporter CTR2 has been identified for the mechanisms of platinum transport, but it functions as an efflux transporter [93]. Knockdown of CTR2 is associated with increased platinum cellular accumulation and efficacy. CTR2 is expressed mainly in lysosomes and late endosomal formations; therefore, regulation of endocytosis is suggested to be involved in cisplatin efflux by CTR2 [125]. There are other transporters, such as OCT1, OCT2 and OCT3, and small GTPases that are believed to serve as regulators of cisplatin trafficking [4].

## 10. Increased Cisplatin Detoxification

Glutathione conjugation to cisplatin mediated by glutathione transferase has been linked to nephrotoxicity and ototoxicity. Cisplatin has a high affinity to glutathione because of its nucleophilic nature. Platinum-glutathione conjugates are subjected to extracellular transport by MRP proteins. Cisplatin sequesters with glutathione S-transferase (GST P1-1) resulted in cisplatin inactivation and inhibition of apoptosis signalling by c-JUN terminal kinase [129]. Similarly, metallothionein (MT) can promote cisplatin diffusion and detoxification. MT are cysteine-rich proteins responsible for metal homeostasis. MT2 overexpression was previously described in bladder carcinomas resistant to platinum [130]. Comparably, in non-small cell lung cancer, MT2 upregulation was evident in patients following platinum therapy as well as in vivo murine models [131]. More recently, MT3 upregulation in neuroblastoma was correlated with refractory mechanisms to cisplatin [132]. Thus, inhibitors of the specific metallothionein may serve to diminish the cisplatin dose while maintaining its cytotoxic and genotoxic effects.

## 11. Epigenetics Changes

As cisplatin primarily targets the DNA, it is predicted that cellular resistance mechanisms can extend to epigenetic regulations. DNA methylation is a key mechanism for acquired cisplatin resistance. Studies using the DNA hypomethylating agent 2-deoxy-5-aza-cytidine revealed over hundred genes that are hypermethylated in platinum-resistant cell lines and could be reactivated via azacytidine [53]. Methylation of the folate-binding gene (FBP) was shown as a mechanism of cisplatin resistance in hepatocellular carcinoma [133]. Epigenetic profiling aided in understanding the molecular landscape of platinum resistance. A study analysing CpG promoter islands methylation in germ cell tumours revealed hypermethylation of key genes, including the stem cell marker *NANOG* and *POU5F1*, to be drivers of platinum resistance [134].

Histone modifications are also described to be a mechanism for cisplatin resistance [51], and post-translational modifications of histones can regulate many of the processes involved in the resistance, including DNA repair effectors, transcription and signalling. It was found that in head and neck squamous cell carcinomas, NFkβ activation can drive chemoresistance. Active NFkβ signalling promotes histone deacetylation and reduces nuclear BRCA1 levels and increases genomic instability [135]. The combination of belinostat, an inhibitor of histone deacetylase, with decitabine increased the expression of epigenetically silenced MLH1 and MAGE-A1 and increased cisplatin sensitivity in ovarian cancer xenografts [136]. In fact, Histone deacetylase inhibitors seem promising clinically to circumvent cisplatin resistance. Cisplatin plus belinostat are in phase I clinical trials in advanced solid tumours [137], as well as panobinostat which is another FDA histone deacetylase inhibitor [138].

## 12. Upregulation of DNA Repair Capacity

The prominent role of DNA repair pathways in overcoming cisplatin toxicity is well established. The majority of cisplatin lesions are recognised by NER and MMR pathways. Therefore, increased expression of NER and MMR genes is a key mechanism for repairing platinum DNA adducts and controlling chemoresistance [58]. ERCC1 expression is associated with platinum resistance and poor survival in ovarian, bladder, oesophageal, head and neck cancers, as well as in NSCLC [87]. Additionally, MMR-related proteins that participate in the recognition of GpG inter-strand adducts, including MSH2 and MLH1, are mutated in some cisplatin refractory tumours [56]. In fact, microsatellite instability or mismatch repair deficiency is a predictor of patient prognosis and chemotherapy response in endometrial and colorectal cancers [139].

Interestingly, a study compared the microsatellite instability status in ovarian cancers between the primary resected tumours and the secondary resected tumours of the same patients. The amplification of 10 microsatellite loci and immunohistochemical detection of hMSH2 and hMLH1 expression in 24 cases of ovarian cancers revealed that all the secondary resected tumours showed microsatellite instability (MSI). Of the 24 primary tumours, 15 had MSS status; however, their residual tumours after 5 or 6 courses of platinum exhibited MSI status through loss of the expression of MLH1 [140].

On the contrary, a similar analysis for MMR markers, including MLH1, was performed on cervical cancer patients pre and post-neoadjuvant chemotherapy and showed no change in microsatellite stability post-chemo exposure [140]. Thus, the role of mismatch repair deficiency in platinum resistance appears to be dependent on the tumour-specific microenvironment.

Another crucial mechanism for repairing cisplatin-induced DNA adducts is translesion synthesis by a group of DNA polymerases belonging to the Y family (Polymerase η (Polη), Polι, Polκ and Rev1) and B families (such as Polζ) of DNA polymerases. Translesion DNA synthesis (TLS) is a tolerance mechanism by which the cells circumvent deleterious double-strand breaks caused by replication stalling [141].

When the replication fork encounters DNA lesions on the leading strand, replication is stalled. While the lagging strand can still go through the replication mechanism, normal base pairing cannot progress, and eventually, replication will be blocked. The cells develop mechanisms to prevent this hazardous form of DNA damage and continue replication. The cells switch to translesion polymerases to insert bases and fill the gap. Yet this is done with low fidelity and thus induces mutations. Polη can bypass several bulky DNA adducts, including cisplatin-GG adducts. Polη deficient cells display more sensitivity to cisplatin treatment compared to proficient cell lines [141,142].

Another TLS polymerase, Polζ, composed of two subunits, Rev3L and Rev7, is implied in bypassing cisplatin-induced intra-strand adducts. Similarly, it was shown in murine studies that Polζ deficient tumours are more sensitive to cisplatin [143]. Knockdown of the catalytic subunit Rev3L desensitised cells to platinum [144]. Importantly, in a cohort of head and neck squamous cell carcinomas with high polη expression showed an association with poor platinum response and worse outcomes for patients [145]. These findings imply that TLS polymerases play a crucial role in bypassing cisplatin lesions. However, the exact signalling by which the cells activate the polymerase of choice is not clear.

## 13. Future Prospective

Cisplatin is one of the most effective chemotherapeutic anticancer agents. Since it acquired FDA approval five decades ago for the treatment of testicular cancer, it remains very effective in combination with bleomycin and etoposides [146,147]. Other tumours remain responsive to cisplatin like head and neck, ovarian and non-small cell lung cancers due to its actions on various cellular components and the activation of multiple pathways for cell killing [128]. Cisplatin also elects complex mechanisms of resistance. Response to cisplatin involves several DNA repair pathways and epigenetic changes, which drive the cells to develop defence mechanisms against the toxic effects of the drug [51]. As a result, tumours recur with profound molecular and genetic changes that favour cell survival, DNA methylation, gene silencing, or activation that inhibits apoptosis [4]. Thus, current new therapeutic regimes combine platinum with other molecular targeted therapies aiming to inhibit resistance mechanisms. The combination of cisplatin with bevacizumab, a vascular endothelial growth factor inhibitor, remains attractive in non-small cell lung cancers as well as cervical cancers [148,149]. Another interesting approach is the combination of cisplatin with olaparib, the first FDA-approved DNA-targeted therapy. A promising clinical study investigated the approach of combining olaparib with cisplatin and irinotecan for pancreatic ductal carcinoma and found a durable clinical response [150]. Additional clinical studies can explore how to maximise the benefit of this strategy. Approaches for combining cisplatin with checkpoint inhibitors also seem attractive.

Cisplatin nephrotoxicity is a challenging obstacle. A recent report showed that co-administration of cilastatin with cisplatin significantly reduced nephrotoxicity in a manner that permits escalating cisplatin dose. Cilastatin acts as a blocker of megalin, which is an endocytic receptor at the apical membrane of the tubular epithelial cells [151]. Several studies have shown that the transporters CTR1 and OCT2 are responsible for cisplatin uptake into cells. The abundant expression of these transporters on renal tubular cells has a major role in kidney toxicity [121]. Thus, developing small molecule inhibitors that inhibit these transporters or compete with cisplatin for the transporter binding site could have clinical significance by reducing nephrotoxicity. The current advent in proteomic and transcriptomic studies is expected to yield new targets that will aid in stratifying patients and when combined with DNA repair defects, should enhance the antitumour effect of cisplatin. At the moment, the challenge remains to find ways to bypass the molecular processes causing cisplatin resistance and with the many targets highlighted herein, it seems that downregulating multiple processes would be required to maximise the anticancer benefits of cisplatin.

## Figures and Tables

**Figure 1 ijms-23-07241-f001:**
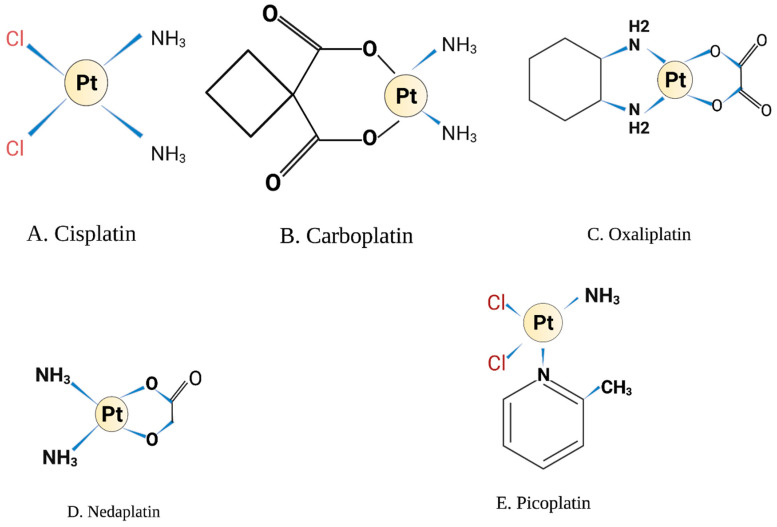
Chemical structure. Illustrations created by BioRender.com (accessed on 8 May 2022).

**Figure 2 ijms-23-07241-f002:**
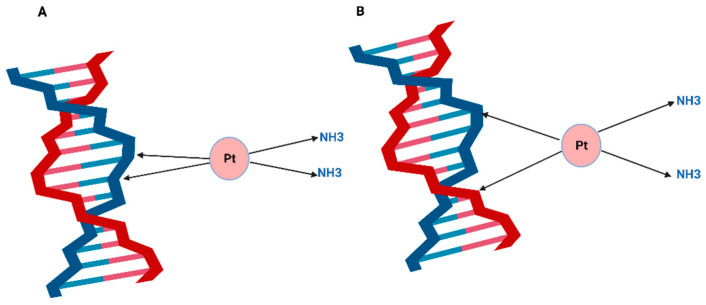
Cisplatin DNA adducts are formed when cisplatin exchange one or two of its chloride molecules for water and bind covalently to the purines at the N7 position to form (**A**) Cisplatin intra-strand adduct or (**B**) Cisplatin inter-strand crosslink. Illustrations created by BioRender.com (accessed on 8 May 2022).

**Table 1 ijms-23-07241-t001:** Recent studies on the combination of platinum derivatives with other drugs.

Drug	Cancer	Outcome	Year	Reference
Gemcitabine and cisplatin plus durvalumab with or without tremelimumab	Advanced biliary tract cancer	Gemcitabine and cisplatin plus durvalumab are still being evaluated. However, Gemcitabine and cisplatin plus immunotherapy showed an acceptable level of safety and are considered a potential effective first-line therapy for advanced biliary tract cancer patients	2022	[19]
Addition of nintedanib or placebo to neoadjuvant gemcitabine and cisplatin	Advanced muscle-invasive bladder cancer	Nintedanib addition was safe but did not show improvement in the pathological response for the targeted bladder cancer patients.	2022	[20]
Xevinapant or placebo plus cisplatin	Advanced squamous cell carcinoma of the head and neck	No results have been released yet	2022	[21]
Pembrolizumab plus a high dose of cisplatin and radiation	Larynx cancer	The therapy was safe and showed effectiveness, but more data and longer-term monitoring are needed	2022	[22]
Stereotactic body radiotherapy and a full dose of cisplatin or carboplatin	Lung cancer	The therapy was effective and safe	2022	[23]
TRC102 in combination with pemetrexed, cisplatin, and radiotherapy.	Lung cancer	The therapy was safe with specific doses of TRC12 (200 mg along with cisplatin)	2022	[24]
Cisplatin coupled with radiation followed by carboplatin/paclitaxel vs carboplatin/paclitaxel	Endometrial carcinoma	Chemoradiotherapy showed some toxicity in comparison to chemotherapy alone	2022	[25]
Liposomal cisplatin versus conventional non-liposomal cisplatin	Lung cancer and squamous cell carcinoma of the head and neck	Liposomal cisplatin showed a significant reduction in toxicity compared to conventional therapy	2018	[26]
Anlotinib, in combination with oxaliplatin and capecitabine	Colorectal adenocarcinoma	Anlotinib combined with capecitabine and oxaliplatin exhibited significant effectiveness as first-line therapy with manageable toxicity	2022	[27]

**Table 2 ijms-23-07241-t002:** Recent clinical trials on Cisplatin.

Molecule	Cancer Type	Status	NCT Number	Year
Atezolizumab, Bevacizumab, Placebo, Cisplatin, Gemcitabine	Biliary Tract Cancer	Phase II	NCT04677504	2021
Pembrolizumab Carboplatin Paclitaxel Placebo for pembrolizumab docetaxel Cisplatin Radiation: External Beam Radiotherapy (EBRT) Cisplatin (as radiosensitiser) Radiation: Brachytherapy	Endometrial Neoplasms	Phase III	NCT04634877	2021
Pembrolizumab Lenvatinib Cisplatin 5-FU Oxaliplatin Leucovorin Levoleucovorin Paclitaxel	Metastatic Esophageal Squamous Cell Carcinoma	Phase III	NCT04949256	2021
Zanidatamab Tislelizumab Trastuzumab Capecitabine Oxaliplatin Cisplatin 5-FluorouracilDiagnostic Test: In situ hybridisation (ISH)-based companion diagnostic assayDiagnostic Test: Immunohistochemistry (IHC)-based companion diagnostic assay	Gastric Neoplasms Gastroesophageal Adenocarcinoma Esophageal Adenocarcinoma	Phase III	NCT05152147	2021
SavolitinibDrug: Osimertinib Pemetrexed Cisplatin Carboplatin	Carcinoma Non-Small-Cell Lung	Phase III	NCT05261399	2022
Olaparib Radiation: Pelvic external beam radiotherapy Cisplatin Durvalumab Medroxyprogesterone Acetate Megestrol Acetate Other: Observation	Endometrial Cancer	Phase II & III	NCT05255653	2022
Nab-paclitaxel Gemcitabine Cisplatin Irinotecan Capecitabine Pembrolizumab Olaparib	Metastatic Pancreatic Cancer	Phase II	NCT04753879	2021
Bintrafusp Alfa Pemetrexed Carboplatin Cisplatin	Locally Advanced Lung Non-Squamous Non-Small Cell Carcinoma. Metastatic Lung Non-Squamous Non-Small Cell Carcinoma. Unresectable Lung Non-Squamous Non-Small Cell Carcinoma	Phase II	NCT04971187	2021
AMG 510 Cisplatin Carboplatin Pemetrexed	Lung Cancer	Phase II	NCT05118854	2022
Cisplatin	SCCHN Squamous Cell Carcinoma	Phase II	NCT04595981	2022
Modified GCN+TTF treatment	Metastatic Pancreatic Cancer. Pancreatic Adenocarcinoma. Metastatic Adenocarcinoma	Phase I & Phase II	NCT04605913	2022
BET Bromodomain Inhibitor ZEN-3694 Cisplatin Etoposide	Advanced NUT Carcinoma Metastatic NUT Carcinoma Unresectable NUT Carcinoma	Phase I & Phase II	NCT05019716	2022
Cisplatin and immunotherapy	Cholangiocarcinoma	Phase I & Phase II	NCT04989218	2022
Sasanlimab Radiation: Stereotactic Body Radiation Therapy Procedure: Radical Cystectomy + pelvic lymph node dissection + urinary diversion	Urothelial Carcinoma Bladder	Phase II	NCT05241340	2022
Zimberelimab Etrumadenant Cisplatin Radiation	Head and Neck Cancer. Squamous Cell Carcinoma of Head and Neck. Oral Cavity Squamous Cell Carcinoma. Oropharynx Squamous Cell Carcinoma. Larynx Cancer. Pharynx Cancer. Hypopharynx Cancer. Hypopharynx Squamous Cell Carcinoma	Phase I	NCT04892875	2022

**Table 3 ijms-23-07241-t003:** Recent clinical trials on cisplatin derivatives.

Molecule	Cancer Type	Status	NCT Number	Year
Carboplatin and with combinations and other drugs versus Sacituzumab Govitecan-hziy	Triple-Negative Breast Cancer PD-L1 Negative	Phase III	NCT05382299	2022
Carboplatin and other drugs versus Sacituzumab govitecan-hziy (SG) and pembrolizumab	Triple-Negative Breast Cancer PD-L1 Positive	Phase III	NCT05382286	2022
Olvi-Vec followed by platinum-doublet chemotherapy (carboplatin or cisplatin) and bevacizumab compared to the Active Comparator Arm with platinum-doublet chemotherapy (carboplatin or cisplatin) and bevacizumab	Platinum-resistant Ovarian Cancer Platinum-refractory Ovarian Cancer Fallopian Tube Cancer Primary Peritoneal Cancer High-grade Serous Ovarian Cancer Endometrioid Ovarian Cancer Ovarian Clear Cell Carcinoma	Phase III	NCT05281471	2022
Pembrolizumab/vibostolimab (MK-7684A) in combination with other drugs including Cisplatin or Carboplatin versus pembrolizumab in combination with other drugs including Cisplatin or Carboplatin	Metastatic Non-Small Cell Lung Cancer	Phase III	NCT05226598	2022
Pembrolizumab (MK-3475) compared to a combination of carboplatin and paclitaxel	Endometrial Neoplasms	Phase III	NCT05173987	2022
Efficacy and safety of Dato-DXd compared with Investigator’s choice chemotherapy such as carboplatin	Breast Cancer	Phase III	NCT05374512	2022
Fixed-dose of pembrolizumab/vibostolimab co-formulation (MK-7684A) with etoposide/platinum chemotherapy (cisplatin, carboplatin, or others) followed by MK-7684A compared to the combination of atezolizumab with etoposide/platinum chemotherapy (cisplatin, carboplatin, or others) followed by atezolizumab	Small Cell Lung Carcinoma	Phase III	NCT05224141	2022
Patritumab Deruxtecan versus Platinum-based chemotherapy	Non-squamous Non-small Cell Lung Cancer EGFR L858R	Phase III	NCT05338970	2022
Pembrolizumab/vibostolimab (MK-7684A) in combination with concurrent chemoradiotherapy including cisplatin or carboplatin followed by pembrolizumab/vibostolimab versus chemoradiotherapy including cisplatin or carboplatin followed by durvalumab	Carcinoma, Non-Small-Cell Lung cancer	Phase III	NCT05298423	2022
Savolitinib in combination with osimertinib versus platinum-based doublet chemotherapy including cisplatin or carboplatin	Carcinoma Non-Small-Cell Lung Phase 3	Phase III	NCT05261399	2022
Different strategies using the following: Olaparib Radiation: Pelvic external beam radiotherapy Chemotherapy, including cisplatin and carboplatin Durvalumab Medroxyprogesterone Acetate Megestrol Acetate Other: Observation	Endometrial Cancer	Phase II & III	NCT05255653	2022
Trials with different combinations of the following drugs: Tucatinib Trastuzumab Bevacizumab Cetuximab Oxaliplatin Leucovorin Levoleucovorin Fluorouracil	Colorectal Neoplasms	Phase III	NCT05253651	2022
Specifying the best therapy among the following: mFOLFOX6 3–6-month CAPOX 3-month mFOLFIRINOX mFOLFOX6 6 month CAPOX 6 month	Colon Cancer	Phase III	NCT05174169	2022
Bemarituzumab combined with oxaliplatin and 5-fluorouracil (5-FU) (mFOLFOX6) versus placebo plus mFOLFOX6	Gastric Cancer	Phase III	NCT05052801	2022
Testing the efficacy of zilovertamab vedotin in combination with other drugs such as oxaliplatin	Diffuse Large B-Cell Lymphoma	Phase II & III	NCT05139017	2022

## Data Availability

College of Health and Life sciences, Hamad Bin Khalifa University, Education city, Doha, Qatar.

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
