# Peer review of "Can Cisplatin Therapy Be Improved? Pathways That Can Be Targeted"

_ijms, 2022, doi:10.3390/ijms23137241_

Round 1

Reviewer 1 Report

The revised manuscript is now can be suitable for the journal. The authors completely restructured the manuscript and important information was added. This manuscript can be accepted after minor modification. Please change Figure 1 with real chemical structures, not a cartoon one, especially for the oxaliplatin as it is an enantiomerically pure compound. 

Author Response

Response 1 : Thank you very much for the reviewer. Figure 1 has been revised and updated with the real chemical structure of the platinum derivatives. And will be uploaded with the manuscript.

Reviewer 2 Report

The manuscript “Can Cisplatin therapy be improved? Pathways that can be targeted” is a detailed systematic presentation of research on cisplatin, which is the "long-lived" antitumor preparation. In addition to characterizing cisplatin, the authors present data on its analogues, which were studied to minimize the toxic effect of platinum compounds while maintaining antitumor activity. The information given in the manuscript and its generalization in the form of a tables will undoubtedly be useful to readers. The information presented in the manuscript on ongoing clinical trials of cisplatin and its derivatives in combination with various drugs will undoubtedly be in demand.

               Having considered the application of cisplatin derivatives in order to find the least toxic variant of platinum compounds, the authors analyze the possibility of using nanoparticles as another approach to reduce the toxic properties of cisplatin and carboplatin. It is useful that the published data are presented in terms of drug interactions with cells both in this section of the review, and in subsequent ones, considering the importance of the mechanisms of penetration of drugs into cells and the ways of self-defense of tumor cells from the toxic effects of chemicals.  

The review analyzes data on the mechanisms of DNA damage by cisplatin and the mechanisms of DNA repair that ensure the survival of tumor cells treated with platinum preparations. The role of defects in DNA repair systems in the development of the disease and the need for their analysis in patients for successful treatment are noted. Authors showed also the whole variety of mechanisms of interaction between cisplatin and the patient's body, including mutations in some genes and immune responses. The section "Cisplatin uptake transporters" contains very interesting information about the involvement of copper transporters in the uptake of platinum into cells, and cisplatin-glutathione interaction.

Overall, this review presents a truly comprehensive analysis of the interaction of cisplatin with tumors, from the molecular level to clinical trial data that show the effectiveness of the drug. The authors used a wide range of publications, reflecting the current state of the study of cisplatin, and the text can serve as a good source of reference information. It should be emphasized that the text is very clear and easy to read, which is important for readers who are not directly involved in the work with cisplatin. In general, the manuscript “Can cisplatin therapy be improved? Pathways That Can Be Influenced" is a complete "state of the art" view of current knowledge of cisplatin toxicity and how to overcome it.

Author Response

We would like to thank the reviewer for his comments. We would like to sincerely appreciate his appraisal of this piece of work.

This manuscript is a resubmission of an earlier submission. The following is a list of the peer review reports and author responses from that submission.

Round 1

Reviewer 1 Report

Dear authors,

The manuscript “Can Cisplatin therapy be improved? Pathways that can be targeted”, for the first glance looks attractive and interesting. However, as you read, you get the feeling that the time machine took you 20 years ago. The review is written mainly on the basis of publications from 2002-2014, and this impression is confirmed on page 6: "A recent report showed that cisplatin sequesters with GST P1-1 lead to cisplatin inactivation and inhibition of apoptosis signaling by c-JUN terminal kinase (Sawers et al., 2014). Indeed, out of a total of 58 references, only 2 were published in 2020, and 5 in 2021, including reference No. 2, representing the work of one of the authors of this review.

Since cisplatin is well established as an anticancer drug, it is under intense research, as reflected by thousands of publications.

PubMed: (mechanisms) AND (cisplatin cytotoxicity) 2,639 results; (cisplatin toxicity) - 23,028 results.

I am forced to conclude that the presented review has no scientific or even cognitive value, despite the intriguing title. Modern knowledge on the cisplatin features is missed. 

I recommend reading the 2021 review to compare its content and the content of your paper:

Paul B Tchounwou et al. Advances in Our Understanding of the Molecular Mechanisms of Action of Cisplatin in Cancer Therapy. J. Exper. Pharmacology 2021:13 303–328.

For the future: I recommend carefully editing the text, and even better - to involve a native English speaker in editing, there are many semantic errors in this manuscript, many punctuation marks are incorrectly placed.

And most importantly, the review of publications should reflect the current state of knowledge of the subject and analyze the main trends of research in the field.

Unfortunately, your review does not meet these requirements and, in my opinion, cannot be published.

Reviewer 2 Report

The presented review on cisplatin and its mechanism of action are one of many already published in different journals. I don't think our field will get any new good knowledge out of this review. Most of the statements are already well known and trivial and some such as "Cisplatin (cis-diamminedichloroplatinum(II)) is the oldest known chemotherapeutic
agent." simply not correct. In my opinion, this review must be rejected from such a high-level journal based on scientific quality.